# Neuregulin-1 prevents death from a normally lethal respiratory viral infection

Syed-Rehan A. Hussain[1,2]*, Michelle Rohlfing[1,2], Jennifer Santoro[1,2], Phylip Chen[3], Kaushik Muralidharan[2], Matthew S. Bochter[1,2], Mark E. Peeples[3,4], Mitchell H. Grayson[1,2,4]*

1 Division of Allergy and Immunology, Nationwide Children's Hospital - The Ohio State University College of Medicine, Columbus, Ohio, United States of America, 2 Center for Clinical and Translational Research, Abigail Wexner Research Institute at Nationwide Children's Hospital, Columbus, Ohio, United States of America, 3 Center for Vaccines and Immunity, Abigail Wexner Research Institute at Nationwide Children's Hospital, Columbus, Ohio, United States of America, 4 Department of Pediatrics, The Ohio State University College of Medicine, Columbus, Ohio, United States of America

* wheeze@allergist.com (MHG); rehan.hussain@nationwidechildrens.org (SRAH)

## Abstract

Respiratory infections with RNA viruses such as respiratory syncytial virus (RSV) and influenza lead to significant morbidity and mortality. Using a natural rodent pathogen, Sendai virus (SeV), which is similar to RSV, mice made atopic with house dust mite survived a normally lethal SeV infection. One protein that we found markedly elevated in the lungs and bronchoalveolar lavage fluid of atopic mice was neuregulin-1 (NRG1). Administration of NRG1 protected naïve (non-atopic) mice from death with both SeV and mouse adapted influenza A virus (IAV). Survival was associated with reduced alveolar epithelium permeability and reduced phosphorylation of mixed lineage kinase domain-like (MLKL) protein indicating inhibition of necroptosis. *In vitro*, treatment of mouse lung epithelial cells with NRG1 inhibited SeV induced necroptosis, and NRG1 administration to differentiated human bronchial epithelial cells infected with RSV reduced transepithelial fluid leak and expression of necroptosis associated genes *RIPK3* and *MLKL*, while regulating genes associated with homeostatic maintenance, suggesting stabilized epithelial integrity. In conclusion, our data demonstrate a unique function of NRG1 in respiratory viral infections by reducing alveolar leak, inhibiting epithelial necroptosis, and promoting homeostatic regulation of airway epithelium, all of which associate with markedly reduced mortality to the respiratory viral insult.

## Author summary

Respiratory viral infections, like those caused by influenza and respiratory syncytial virus (RSV), have significant impact leading to increased mortality and morbidity, especially in the very young and elderly. Interestingly, mice that have been made allergic to house dust mite are resistant to normally lethal doses of

**Data availability statement:** All data underlying our findings are fully available without restrictions. RNA seq data, relevant to the submitted manuscript, have been submitted to GEO repository (accession number GSE281143) and can be accessed at https://www.ncbi.nlm.nih.gov/geo/query/acc.cgi?acc=GSE281143

**Funding:** This work was supported by National Institutes of Health (NIH) grants (R01HL087778 and R01AI171027 (to MHG); R01 AI093848 and U19AI42733 (to MEP)), the Robert F Wolfe & Edgar T Wolfe Foundation and The Abigail Wexner Research Institute at Nationwide Children's Hospital (both to MHG). The funders had no role in study design, data collection and analysis, decision to publish, or preparation of the manuscript.

**Competing interests:** I have read the journal's policy and the authors of this manuscript have the following competing interests: MHG is the Editor-in-Chief of Annals of Allergy, Asthma & Immunology, serves on an advisory board for Bayer and has stock options in Invirsa, Inc. MHG and SRAH are co-inventors on a patent application (US2022/078494) on the use of Neuregulin for protection against respiratory viral infection (filed by the Research Institute at Nationwide Children's Hospital).

influenza and murine parainfluenza virus type 1 (Sendai virus, SeV). We found that the protein neuregulin-1 (NRG-1) is elevated in the lungs and airways of these allergic mice. Instilling NRG-1 into the airways of mice before they are infected with a normally lethal dose of SeV or influenza led to protection from death. Administering NRG-1 to human bronchial epithelial cells (hBEC) grown in culture reduced RSV replication, suggesting this protection may apply to humans. Further, we found that NRG-1 reduced fluid leak into the airways of infected mice and prevented fluid leak in the hBEC cultures infected with RSV. This reduction in fluid leak was associated with a reduced level of necroptosis both in the mice and hBEC cultures. Together, we conclude that administration of NRG-1 before a normally lethal respiratory viral infection prevents epithelial necroptosis, reduces fluid leak into the airways, and subsequently increases survival.

## Introduction

Respiratory viral infections with negative-strand RNA viruses, such as influenza virus (IAV), respiratory syncytial virus (RSV) and parainfluenza viruses (PIV) are a major cause of morbidity and mortality [1,2]. Several groups have demonstrated pre-existing atopy protects from mortality to IAV in mouse models [3–5]. Human data supports this relationship – in the 2009 H1N1 IAV pandemic patients with asthma hospitalized with IAV had less severe outcomes, while a recent meta-analysis concluded patients with asthma had a significantly lower mortality to COVID-19, and the presence of food allergies was reported to have reduced the likelihood of being infected with SARS-CoV-2 [6–10]. The mechanistic explanation for why pre-existing atopy would be protective is not completely understood, although various explanations have been provided for the increased survival in animal models including activation of NK cells, induction of Type III interferon and expression of TGFβ [3–5].

Similar to IAV, RSV is a major respiratory pathogen, especially for infants and the elderly. In the United States annually there are on average 2.1 million outpatient visits and 58,000 hospitalizations for RSV in children under 5 years of age [11,12]. Globally over 200,000 children under 5 years of age die annually of RSV infection [13]. In those 65 years of age or older, RSV accounts for an average of 177,000 hospitalizations and 14,000 deaths annually, with similar mortality rates to IAV [1,14]. Reducing severity of an RSV infection has potential for significant clinical impact by reducing mortality in infants and adults.

Our high-fidelity model of respiratory viral infection utilizes a natural rodent pathogen, Sendai virus (SeV), a negative single-strand RNA respiratory virus - similar to RSV. SeV faithfully replicates in mouse airway epithelial cells and, unlike RSV, in mice causes severe disease (significant weight loss and even death) [15–18]. Infection with SeV at $2\times10^5$ pfu leads to acute bronchiolitis, which can be lethal in up to 20% of infected mice, followed by a chronic inflammatory response associated with

airway hyper-reactivity and mucous cell metaplasia [19]. These are the cardinal features of human paramyxoviral (PIV), pneumoviral (RSV), and orthomyxoviral (IAV) infection, and we have used this model to identify a novel immune axis, which we have validated in humans [20,21]. Furthermore, a log fold higher dose of SeV (2x10$^6$ pfu) is uniformly lethal to C57BL6 mice [22].

Using this model, we recently defined a mechanistic pathway that explains how pre-existing atopy could prevent development of post-viral airway disease [15]. During this study we noted that SeV infected atopic mice lost less weight than non-atopic mice, suggesting a possible survival benefit of atopy. In the current report we demonstrate that atopy protects from SeV induced mortality similar to previous IAV studies [3–5]. Unlike the IAV studies, we now demonstrate that atopic mice have increased lung and airway levels of neuregulin 1 (NRG1), a cytokine of the epidermal growth factor family, that interacts with ErbB receptor tyrosine kinases [23].

Importantly, exogenously administered NRG1 in non-atopic mice significantly reduced mortality to both SeV and IAV, an effect associated with reduced extravasation of fluid into the airways and reduced necroptosis. Finally, we demonstrate that following RSV infection, NRG1 treatment stabilizes the human airway epithelium by improving epithelial proliferation and repair processes and regulates the expression of genes known to play a role in maintaining homeostasis and causing necroptosis.

## Results & discussion

### NRG1 is increased in atopic mice and prevents death from a respiratory viral infection

To model the impact of the atopic state in preventing severe disease from respiratory viral infection, we made C57BL6 mice atopic by sensitizing and challenging them with house dust mite antigen (HDM (atopic)) or PBS (non-atopic control; NA). We previously demonstrated that making mice atopic (with HDM) and then inoculating with 2x10$^5$ pfu ("regular" dose) SeV i.n. 3d after the last HDM challenge attenuated weight loss, reduced SeV titer, and inhibited SeV induced airway hyperreactivity. This prevention of post-viral disease was IL-4 dependent and required the presence of PMN in the atopic mice [15]. Given that atopic mice had suppressed respiratory viral induced disease with the regular dose of SeV, we examined the impact of pre-existing atopy on a normally lethal dose of SeV (2x10$^6$ pfu; "high" dose). Interestingly, like with IAV infection in mouse models [3–5] all mice made atopic before SeV infection survived the high dose of viral inoculation, while all NA mice did not (Fig 1A). Survival in atopic mice was associated with a small (~1/2 log) but significant reduction in viral titer; however, this titer was still higher than that seen in NA mice infected with regular dose SeV, strongly suggesting that viral titer alone is not the reason for the protection from mortality (Fig 1B).

Since we previously demonstrated a critical role for CD11c$^+$ cells in the immune axis linking SeV to post-viral airway disease [15,17,18], we posited that some of the protection against lethal SeV infection could be due to CD11c$^+$ cells from atopic mice. We analyzed the trancriptional changes in CD11c$^+$ cells that occur as a result of making mice atopic. CD11c$^+$ cells were isolated from atopic and NA mouse lung and RNA-seq performed. While there were several dysregulated genes, one gene product whose expression was increased several-fold in atopic CD11c$^+$ cells was neuregulin-1 (NRG1; Fig 1C, S1 Table).

NRG1, a 44-kD glycoprotein, is a cytokine of the epidermal growth factor family and is expressed in 14 isoforms due to alternate splicing or use of alternate promoters. All isoforms contain either an alpha or beta variant epidermal growth factor (EGF)-like domain at their C-terminus that binds to ErbB receptor tyrosine kinases (ErbB2-ErbB4). The EGF-like motif is sufficient for most biological effects of the full-length ErbB proteins [24,25]. NRG1 expression and exogenous delivery have been shown to be beneficial in coronary heart disease and survival models of viral cardiomyopathy infected with coxsackievirus [26–28] but potentially detrimental in hepatitis C viral infection [29]. Moreover, NRG1 has been reported to be one of the regulators of goblet cell formation in human airway epithelial cells and NRG1 was noted to be increased in the airways of mice sensitized and challenged to ovalbumin [30]. More recently it was shown that NRG1 had an anti-inflammatory effect in a HDM induced atopic dermatitis model [31]. Therefore, based on finding NRG1 message elevated,

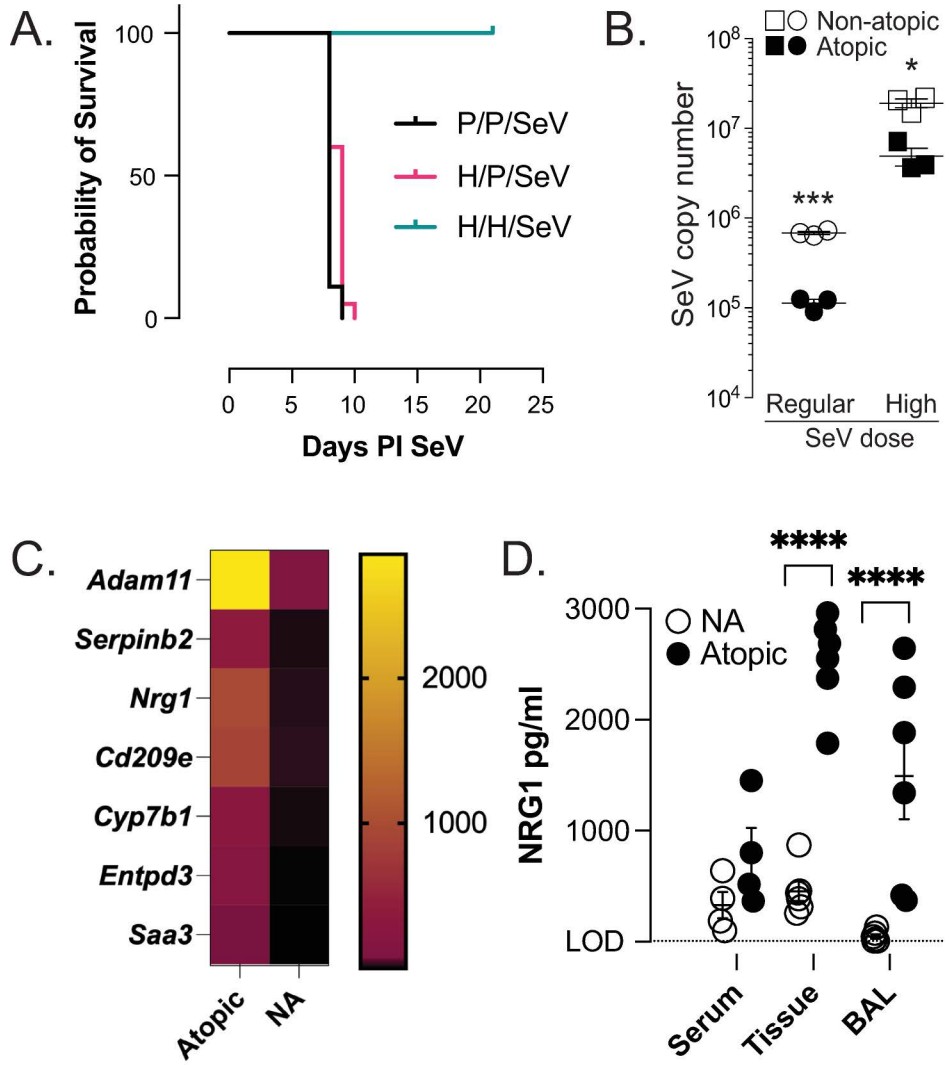

**Fig 1. Atopy prevents lethal SeV infection. (A)** Mice sensitized and challenged with house dust mite extract ("H/H/SeV", atopic) are protected from mortality to high dose (2x10^6 pfu) SeV infection, while NA mice (i.e., only sensitized ("H/P/SeV") or neither sensitized or challenged ("P/P/SeV")) all succumbed to the viral insult ($p < 0.0001$ H/H/SeV versus each of the other NA groups; p = 0.016 P/P/SeV vs H/P/SeV; Mantel-Cox; P/P/SeV n = 9; H/P/SeV and H/H/SeV n = 20 combined from three separate experiments). PI = post inoculation. **(B)** Peak SeV titers (day 5 PI; both at regular (2x10^5 pfu) and high dose (2x10^6 pfu) inoculation) are reduced in atopic mice compared to NA mice, N = 3 per group. **(C)** Transcriptomic (RNAseq) comparison between FACS isolated lung CD11c+ cells from atopic and NA mice identifies several disparately expressed gene products, including *Nrg1* (n = 4 per group). Selected gene products shown. **(D)** NRG1 protein is markedly increased in atopic mouse lung ("tissue") and BAL, N = 6 per group, *p < 0.05, ***p < 0.001.

we measured NRG1 protein levels and found them significantly elevated in the lungs and airways of atopic mice (Fig 1D). We next assessed whether NRG1 could provide a protective advantage with a respiratory viral infection.

We administered NRG1 (10–1000 ng daily) i.n for 5 days to naïve mice before infecting with high dose SeV and determining mortality. Survival increased with as little as 1 ng NRG1 daily, with the effect plateauing at 500–1000 ng (Fig 2A). Given the equal effect of the 500 and 1000 ng dose, we focused on the 500 ng dose for all subsequent *in vivo* experiments. NRG1 administered in a similar manner provided equivalent protection against a normally lethal dose of IAV (Fig 2B), suggesting that NRG1 mediated survival is not limited to a single viral species. Although there was 75% survival in mice receiving 500 ng of NRG1 we did not see a significant reduction in SeV copy numbers (Fig 2C). The effect on

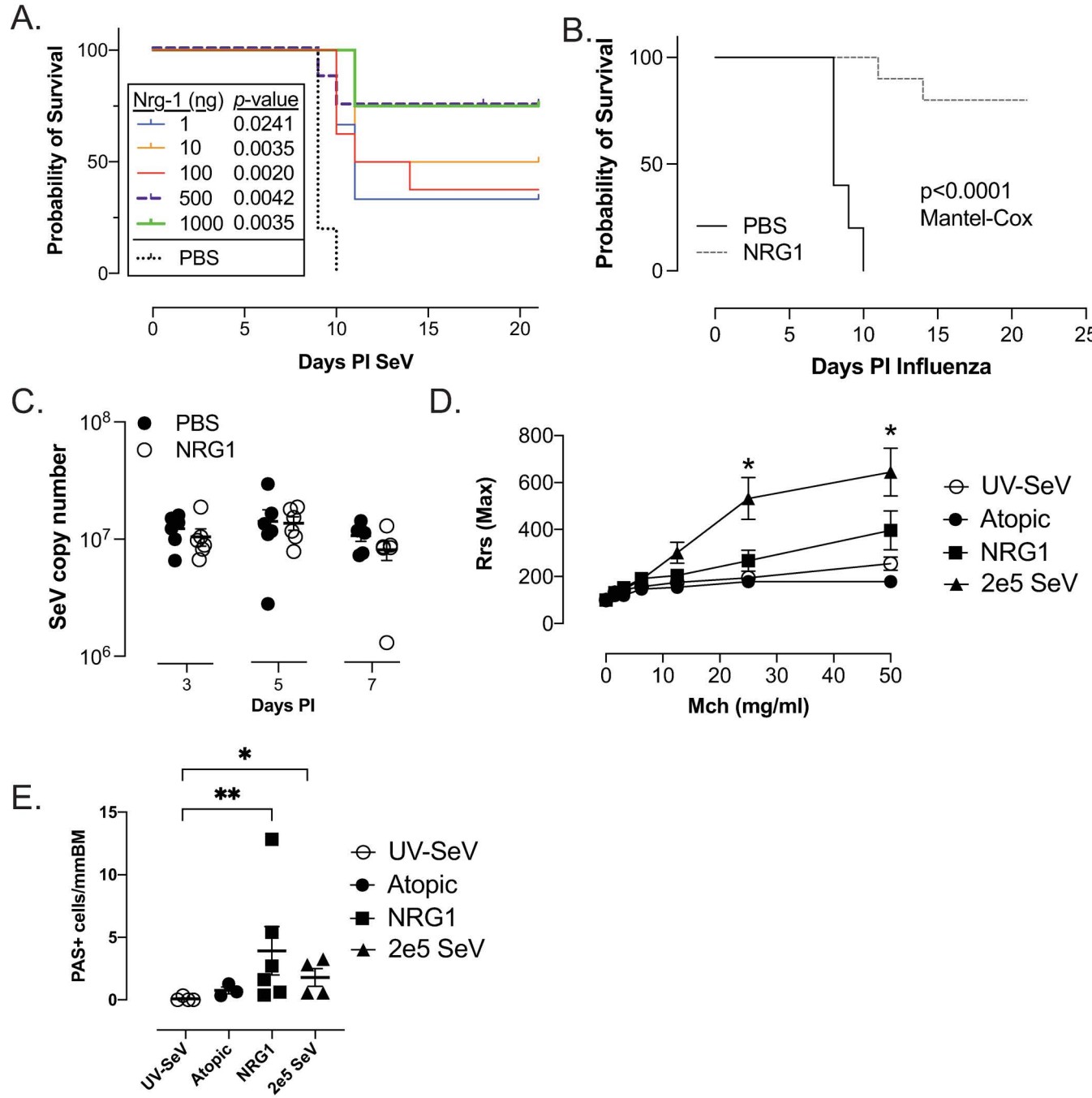

**Fig 2. NRG1 is sufficient to reduce mortality to respiratory viral infections. (A)** NRG1(1 to 1000ng) i.n. (in 30μL) given daily to naïve mice for 5d before inoculation with high dose SeV reduces viral mortality; n = 4 per group (1ng, 10ng 1000ng & PBS) n = 8 per group (100 ng and 500 ng). **(B)** NRG1 (500ng) i.n. (in 30μL) given daily to naïve mice for 5d before infection with 3000 pfu of influenza A (IAV) virus PR8 reduces mortality; n = 5 PBS and n = 10 NRG1 treated (all received IAV). **(C)** SeV titers (high dose) are not reduced in NRG1 treated compared to PBS treated mice. N = 6 per group, per day. **(D)** NRG1 partially reduces post-viral airway disease. NRG1 (500 ng) delivered i.n. for 5 days (d-4 to d0; "NRG1") before inoculation with high dose (2x10⁶ pfu) SeV led to a slight (but not significant) increase in airway hyper-reactivity but did significantly increase **(E)** PAS⁺ cells (mucous cell metaplasia) 21 d post infection when compared to ultraviolet light inactivated SeV (UV-SeV) controls. N ≥ 3 per group; for comparison "2e5 SeV" group (regular dose SeV (2x10⁵ pfu) in NA mice) and atopic mice with 2x10⁶ pfu SeV are shown. *p < 0.05, **p < 0.01.

post-viral airway disease was more nuanced with mucus cell metaplasia (MCM) increasing as previously reported [30], while AHR was blunted but not prevented (Fig 2D and 2E); these data support the contention that the survival mechanism is distinct from that of post-viral airway disease. We found that NRG1 administration to mice infected with high dose SeV had no effect on granulocyte numbers in the lung during the acute infection but did lead to a marked reduction in inflammation by day 21 PI SeV (even less than that seen in regular dose infected non-atopic mice; S1 Fig).

## NRG1 reduces airway epithelial leak *in vivo* and necroptosis *in vitro*

Pulmonary microvasculature and alveolar epithelium form a barrier that functions to maintain lung fluid balance. Injury to lung can disrupt this barrier function leading to further exacerbation of lung injury [32–34]. We hypothesized that one factor leading to death from a severe respiratory viral infection might be an increase in lung epithelial barrier permeability leading to increased fluid leak into the airway. If this hypothesis were true, we would expect NRG1 to reduce airway leakage during SeV infection. We administered 500 ng NRG1 or PBS for 5 days before infection first with SeV ($2\times10^5$ pfu, the dose at which mice survive SeV, so that the PBS treated animals would still be alive) or UV-SeV as control and administered Evans Blue dye (EBD) i.v. on day 8 post infection to measure vascular and airway leak. We have previously demonstrated that day 8 post infection is the peak of vascular leak in the SeV model [35]. As hypothesized, there was an increase in lung vascular leak (increased EBD in the lung) and epithelial permeability/airway leak (as evidenced by EBD in the bronchoalveolar lavage, BAL) in the PBS treated and SeV infected animals (Fig 3A). Administration of NRG1 prior to viral infection, however, significantly reduced EBD in the BAL (Fig 3A, left panel) but not lung (Fig 3A, middle panel), suggesting improved epithelial membrane integrity, but limited effect on endothelial (i.e., vascular) cell leak into the lung. The ratio of EBD in the BAL to that in the lung demonstrates the marked reduction in airway leak with NRG1 treatment (Fig 3A, right panel). We selected a non-lethal SeV dose because otherwise on day 8 post infection there would have been significant mortality with high dose SeV ($2\times10^6$ pfu) in the PBS treated animals. However, to determine if NRG1 was effective in reducing airway permeability at high dose SeV we measured EBD levels at day 5 PI SeV ($2\times10^6$ pfu) in the BAL and lung (Fig 3B, left and middle panel), since this was a timepoint where PBS treated animals would still be alive. EBD levels at this earlier timepoint were not different in the lung but did demonstrate a reduction in BAL of NRG1 treated mice (Fig 3B, left panel). Calculating the percent EBD in the BAL versus the lung, we again saw a statistical reduction in airway EBD with NRG1 administration (Fig 3B, right panel). Thus, even at a suboptimal timepoint for airway leak, with high dose SeV infection NRG1 appears to limit airway but not lung fluid leak. Further studies are needed to determine how NRG1 reduces airway fluid leak, but our data supports the idea that barrier function relates to mortality, with increased airway leak (fluid in the BAL) associating with death, while treatment with NRG1 inhibited airway leak and is associated with survival. Similar to NRG1 treatment, atopic mice also demonstrated reduced airway leak when infected with either low or high dose SeV (S2 Fig).

Recently it was shown that RSV can induce necroptosis (a regulated form of necrosis), which unlike apoptosis, elicits pro-inflammatory responses that can delay viral clearance and alter epithelial permeability [36,37]. Furthermore, necroptosis has been associated with severe IAV infections with activation of receptor-interacting protein kinase 3 (RIPK3) being a key component inducing necroptosis [38,39]. In fact, recently RIPK3 blockade has been shown to inhibit IAV-triggered necroptosis in alveolar epithelial cells with reduced lethality in an IAV mouse model [39]. The key signaling events in necroptosis are the interaction of RIPK3 with receptor-interacting protein kinase 1 (RIPK1) and phosphorylation of executioner protein mixed lineage kinase domain-like protein (MLKL) [40], a process that is caspase 3 independent, demonstrated by reduced levels of active (cleaved) caspase 3 (CC3) in cells undergoing necroptosis [37,41].

We hypothesized that epithelial cell apoptosis or necroptosis could lead to a loss of barrier function, and thus, we examined the effect of NRG1 on these mechanisms. In the small airways of NRG1 treated SeV infected mice the number of cells expressing CC3 were increased over the number found in PBS treated SeV infected mice alone (Fig 4A, 4B). As mentioned, CC3 is reported to be low or inhibited in cells undergoing necroptosis following viral infection, suggesting

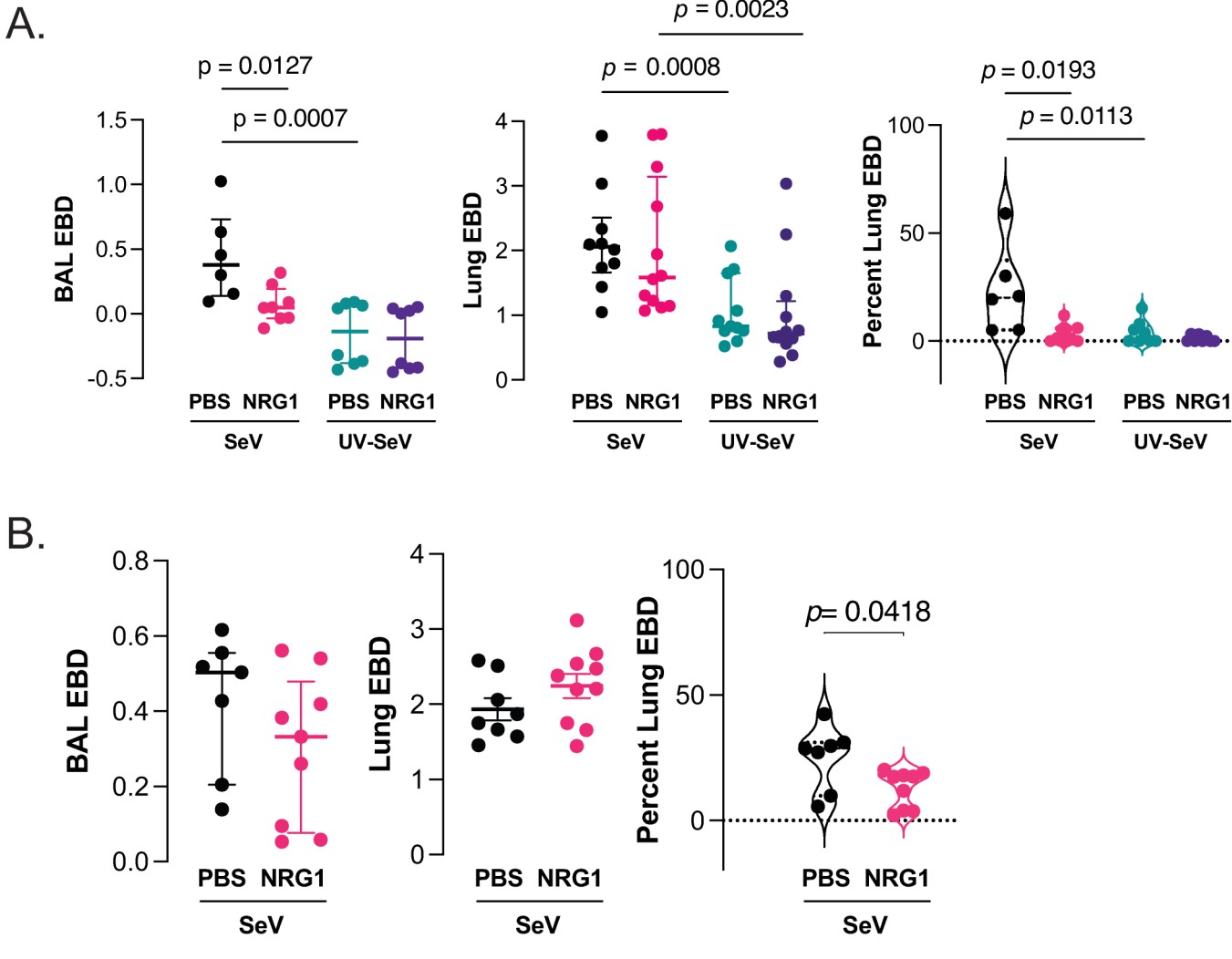

**Fig 3. NRG1 reduces airway fluid leak in vivo. (A)** BAL and lung EBD extravasation at d8 PI regular dose SeV (2x10^5 pfu) or UV-SeV. Levels of EBD in BAL (left), lung (middle), and ratio of EBD in the BAL to that in the lung for corresponding treatments (right) show reduced airway (BAL) but not vascular (lung) leakage in NRG1 treated mice. **(B)** BAL and lung EBD extravasation at d5 PI high dose SeV (2x10^6 pfu) as in (A). For (A) and (B) median±IQR shown, Mann-Whitney U test, n≥6.

that NRG1 was reducing necroptosis in the airway epithelial cells [33,36,41]. The final step in the necroptosis pathway involves phosphorylation of executioner protein MLKL, and therefore, we examined whether MLKL expression and phosphorylation was affected by NRG1. Mice given NRG1 (500ng) for 5 days before SeV infection demonstrated significantly reduced MLKL phosphorylation on day 3 post SeV infection compared to the lungs of mice that received only high dose SeV (Fig 4C, 4D).

Since our *in vivo* data suggested a possible effect of NRG1 on airway epithelial viability, which appeared to be related to reduced necroptosis, we looked *in vitro* to determine if NRG1 had an effect on apoptosis and/or necroptosis in airway epithelial cells. We treated LET1 cells (mouse airway epithelial cells) with NRG1 followed by SeV infection and used a real-time non-lytic assay to differentiate secondary necrosis occurring during late-phase apoptosis (Fig 5). Although this assay does not specifically determine necroptosis; necrosis, necroptosis and secondary necrosis are all characterized by

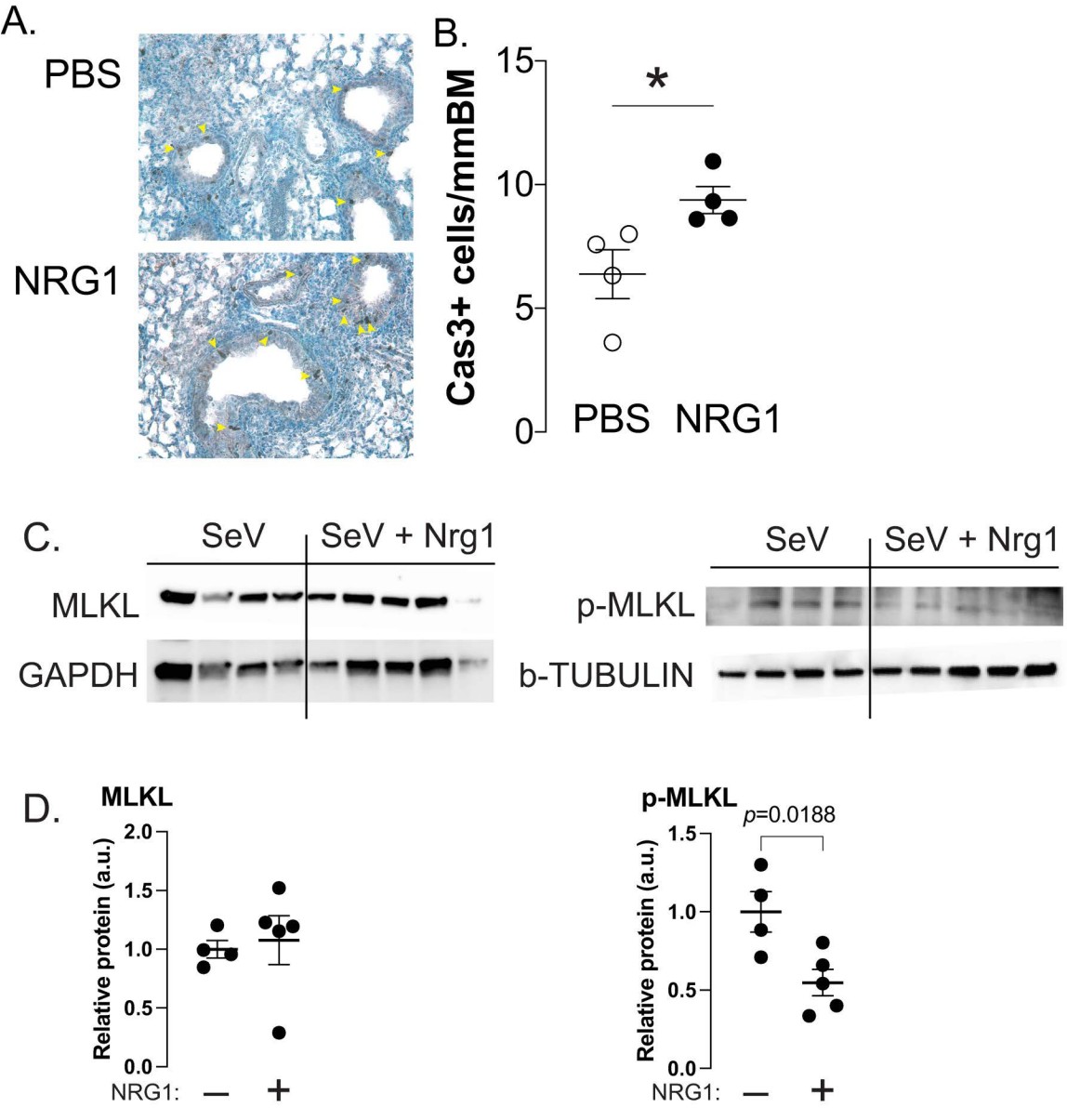

**Fig 4. Active caspase-3 is increased while p-MLKL is reduced in NRG1 treated SeV infected mouse lungs. (A)** Representative images (at 20x magnification) of lung immunohistochemistry showing active caspase-3 stained epithelial cells (arrow heads) in PBS (top panel) and NRG1 treated (bottom panel) (500 ng, daily day-4 to day 0) at day 5 post inoculation SeV. **(B)** Quantification of (A) expressed per mmBM. *p < 0.05. **(C)** Immunoblot of total MLKL and GAPDH (left panel) and p-MLKL and beta-TUBULIN (right panel) from whole lungs after treatment with NRG1 for 5 days and high dose SeV infection. Lungs are from day 3 post infection. **(D)** Quantification of the data in (C). n ≥ 4 per group.

same sequence of cellular/sub-cellular events [42]. Therefore, this assay along with our other observations more specific to necroptosis (i.e., reduced p-MLKL and increased CC3 in NRG1 treatment groups – Fig 4) suggest that NRG1 treatment of airway epithelial cells causes a significant inhibition of necroptosis in cells infected with SeV (Fig 5A). Apoptosis also was reduced, but to a lesser extent (Fig 5B). Together, these data suggest NRG1 regulates viral induced epithelial membrane disruption and could explain reduced airway leak and increased survival seen in our model.

### NRG1 manages the integrity of human airway epithelium following RSV infection

To determine if NRG1 has a direct effect on respiratory viral replication, we utilized an *ex vivo* human airway epithelial cell culture system. Well-differentiated human bronchial epithelial cell cultures (hBEC) growing at the air-liquid interface were treated on the basolateral side with NRG1 (10 ng, 50 ng, 100 ng) in 500 μl media 5 and 3 days before, as well as during inoculation with 4,000 pfu of rgRSV (RSV expressing GFP). As can be seen in Fig 6(A and B), 48h after inoculation GFP intensity was reduced (which correlates with RSV replication) in the NRG1-treated hBEC. Interestingly, after application of NRG1 to the apical side of the hBECs, the reduction in GFP intensity was less than that seen when applying NRG1 to the basolateral side (Fig 6C); these results are similar to what we observed when giving NRG1 i.n. to mice (which more likely mimics an apical not basolateral administration).

Furthermore, supporting our *in vivo* studies that suggested NRG1 reduces epithelial permeability, RSV infected hBEC cultures treated with NRG1 on either the apical or basolateral side demonstrated less FITC-dextran leakage compared to cell cultures treated with RSV alone, although only apical administration was statistically significantly less than RSV alone (Fig 6D). In order to determine if the *in vitro* epithelial leak post RSV inoculation was due to necroptosis, we treated hBEC with the known necroptosis inhibitor necrosulfonamide (NSA) and determined FITC-dextran leak after RSV infection. As shown in Fig 6E treating hBEC with NSA significantly reduced the amount of FITC-dextran that crossed the epithelial barrier, suggesting that RSV induced necroptosis of hBEC leads to increased epithelial leak. Thus, these results support our *in vivo* data that NRG1 inhibits necroptosis, which then reduces the permeability of airway epithelium.

### NRG1 promotes proliferation and modulates homeostasis and necroptosis related genes in hBEC

Since NRG1 has been reported to play a role in maintaining epithelial differentiation and promoting epithelial cell proliferation and repair [30,43–46], we examined the effect of NRG1 on proliferation in RSV-infected hBEC. NRG1 pretreatment led to increased thickness of the epithelial cell layer in RSV-infected hBEC (Fig 6F, left & right panel), suggesting that, similar to other epithelial injury models [47], NRG1 supports epithelial cell proliferation and repair processes during a viral infection.

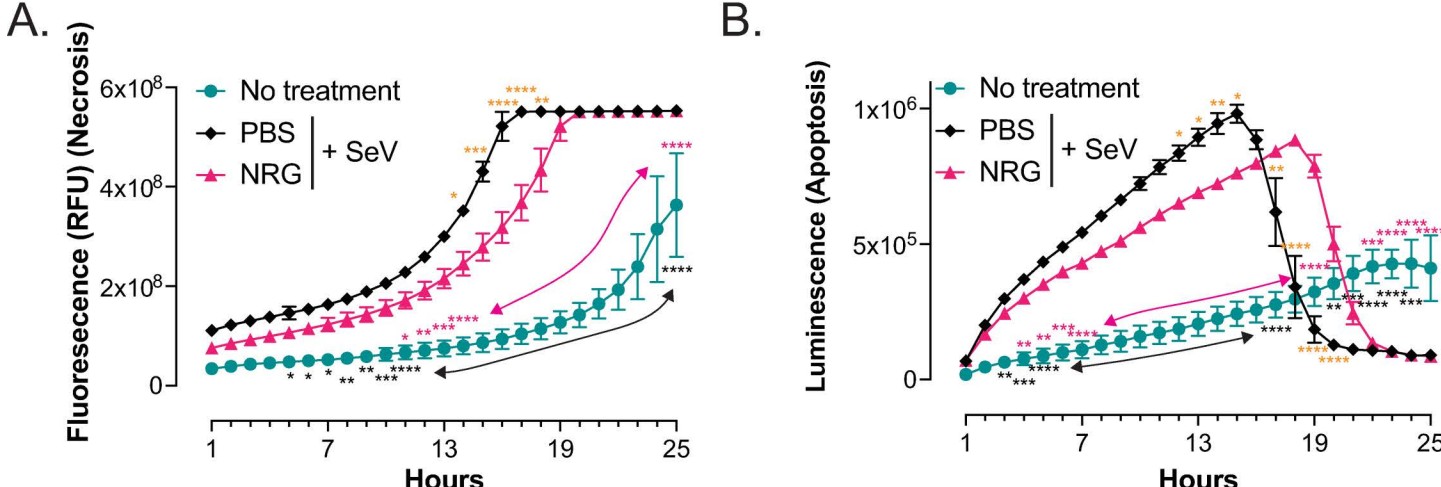

**Fig 5. NRG1 reduces necrosis in mouse airway epithelial cells *in vitro*.** NRG1 treatment reduces (**A**) secondary necrosis and (**B**) apoptosis in LET1 cells infected with SeV (9000 pfu). Data demonstrate necroptosis/apoptosis over 25h of incubation with non-lytic real-time apoptosis and necrosis assay. N = 4; *p < 0.05, **p < 0.01, ****p < 0.0001; orange asterisks compare PBS and NRG1 treated SeV infected cells, pink asterisks compare no treatment with NRG1 treated SeV infected cells, and black asterisks compare PBS treated SeV infected cells with no treatment. Arrows stretch over datapoints with p < 0.0001.

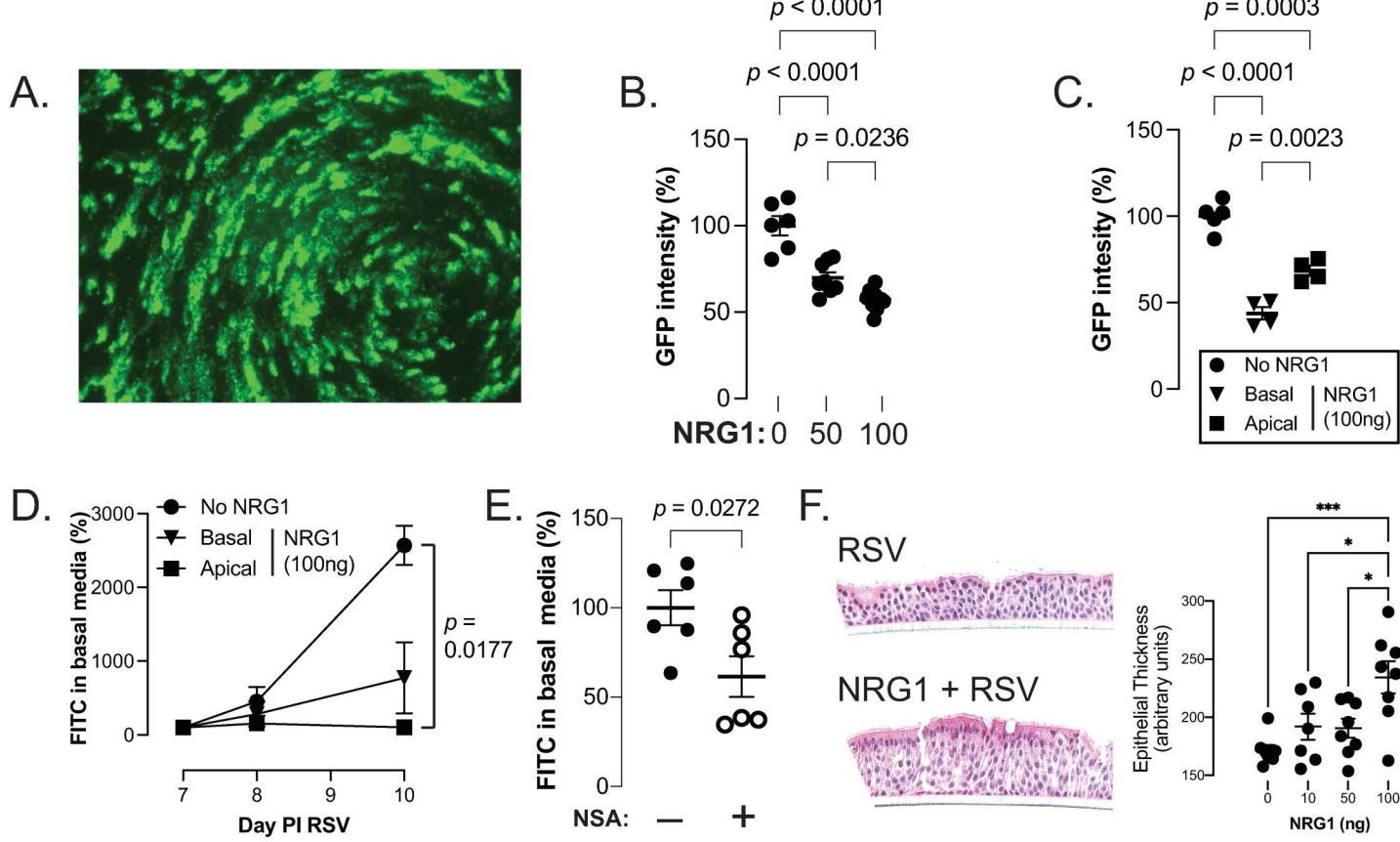

**Fig 6. NRG1 treatment of human epithelium reduces viral replication and epithelial leak, while increasing epithelial layer thickness. (A)** Representative image of hBEC culture 48h post inoculation (PI) with rgRSV (4000 pfu). **(B)** Quantification of GFP expression (measured fluorescence) in hBEC cell cultures treated with or without NRG1 (50 and 100 ng) on the basal side of the culture. GFP expression normalized to the mean of rgRSV infected but NRG1 untreated hBEC. n ≥ 4 separate experiments. **(C)** NRG1 (100ng) applied to the basal media inhibits rgRSV replication significantly more than when applied to the apical side of the transwell. GFP expression analyzed as in (A), n ≥ 4 separate experiments. **(D)** Movement of FITC-dextran from the apical side of the transwell to the basal side is reduced greater in hBEC pretreated for 5 days with 100ng of NRG1 on the apical compared to basal side before apical inoculation with RSV (4000 pfu). Percent FITC in basal media is relative to fluorescence on day 7 PI RSV of each treatment group, n = 3 per group. **(E)** Necroptosis inhibitor, NSA, reduces epithelial leak in cultures on day 9 post RSV (4000 pfu) inoculation. Percent FITC in basal media is relative to the average of the day 9 RSV group, n = 6 **(F)** NRG1 treatment of RSV infected hBEC in a dose-dependent fashion increases epithelial thickness as demonstrated with representative H & E staining (20x, left panel, 100ng NRG1) of paraffin fixed sections of hBEC cultures 48 h after inoculation with RSV (4000 pfu) with or without NRG1 treatment as in (B). Quantification of epithelial thickening (right panel) with ImageJ software. n ≥ 4 per group. *p < 0.05, ***p < 0.001.

Given the evidence for NRG1 inhibiting necroptosis in the mouse (*in vivo* and *in vitro*), we next examined whether NRG1 could inhibit necroptosis gene expression in hBEC cells 72h post RSV infection. As shown in Fig 7A, while expression of *RIPK1*, *RIPK3*, and *MLKL* all increased with RSV infection in hBEC (compared to NRG1 alone), NRG1 addition to RSV infected hBEC cells significantly reduced expression of *RIPK3* and *MLKL* (the two final proteins in the necroptosis pathway) supporting the idea that NRG1 downregulates viral induced necroptosis. Moreover, the lack of a significant reduction in *RIPK1* is consistent with the IAV study showing that RIPK1 activity was not required for RIPK3 kinase dependent necroptosis [39].

NRG1 has been reported to play a critical role in maintaining homeostasis by reducing cellular and mitochondrial stress [48–50]. Therefore, we next wanted to assess if NRG1 treatment affected epithelial homeostasis. Using a custom

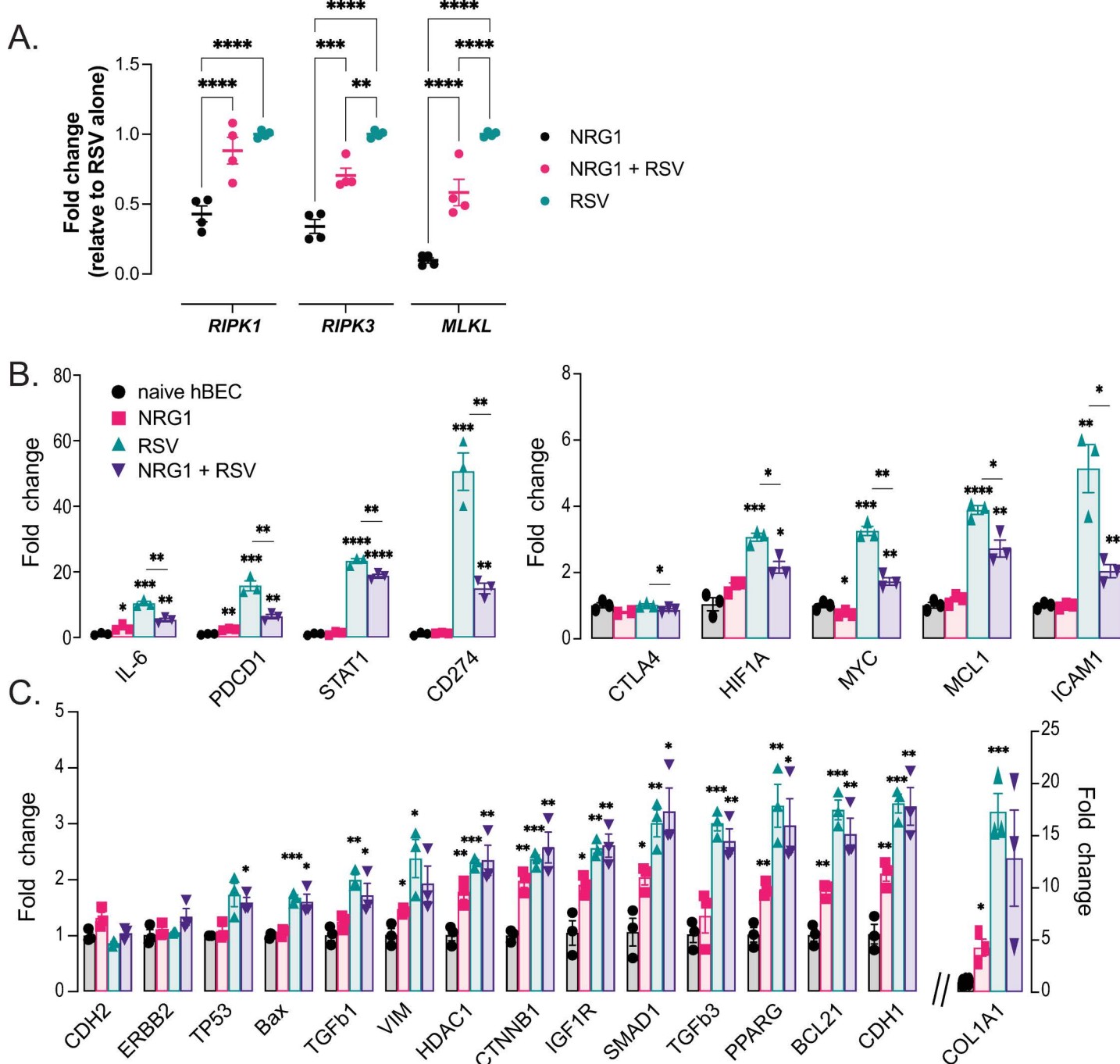

**Fig 7. Regulation of expression of necroptosis and homeostatic genes in human airway epithelium.** Transcriptomic analysis of hBEC cultures treated with NRG1 (100 ng) on the basolateral side for 5 days and inoculated with RSV (4000 pfu). **(A)** RNA isolated 72h PI RSV shows expression of genes involved in necroptosis (normalized to *MRPL13* and relative to RSV). n = 4 per group. **(B)** Transcripts that were reduced with NRG1 treatment compared to levels found in RSV infected cells (left & right panel, note different Y-axis scales). **(C)** Transcripts with low level expression that show small but significant change in expression relative to naïve control with NRG1 alone or genes significantly increased with RSV but whose expression levels were not affected by NRG1. *p < 0.05, **p < 0.01, ***p < 0.001, ****p < 0.0001, n = 3.

**Table 1. Genes assessed in human airway epithelium expression analysis.**

| Gene | Potential functions | Reference |
|---|---|---|
| BAX | Pro-apoptotic-- activation cause mitochondrial dysfunction. | [53] |
| BCL2L1 | Anti-apoptotic proteins that promote cell survival | [53] |
| CDH1 | Epithelial marker role in Cell-cell adhesion and epithelial cell proliferation | [54] |
| CDH2 | Augmented expression in Epithelial to Mesenchymal Transition (EMT) | [55] |
| COL1A1 | Role in EMT | [56] |
| CTLA4 | May regulate PD-L1 expression in non-small cell lung cancer | [57] |
| CTNNB1 | Function in cell adhesion and maintaining lung epithelial progenitors | [58] |
| ERBB2 | Receptor for NRG1 forms dimer with ErBB3 or ERBB4 | [23] |
| HDAC1 | Deacetylation of proteins; Inhibits respiratory viral infection in epithelial cells. | [59] |
| HIF1A | Transcription factor, increased in viral infections indicating mitochondrial damage effecting epithelial repair. Further, HIF1a has been shown to increase epithelial permeability and RIP-induced necroptosis. | [60,61]; [51,62] |
| ICAM1 | Facilitates RSV entry and infection of epithelial cells | [63] |
| IGF1R | Serve as entry receptor for RSV by binding to pre-fusion RSV-F glycoprotein; supports bronchiolar epithelial regeneration. | [64,65] |
| IL6 | Pro- and anti-inflammatory cytokine | [66,67] |
| MCL1 | Anti-apoptotic protein increased in RSV infection and promote viral latency and persistent infection. | [68,69]. |
| MYC | Transcription factor in human airway basal cells and activation has been shown to provide support for respiratory virus replication including IAV and adenovirus. | [70,71] |
| PDCD1 | Encodes PD-1, a receptor for PD-L1; effects inflammatory cell activation and mitochondrial dysfunction. | [72,73] |
| CD274 (PD-L1) | Ligand for PD-1 increased expression in lung epithelial cells in RSV infection. May facilitate antiviral immune response. Increased expression on lung epithelial cells, has been shown to be increased with necroptosis in certain disease condition | [52,74] |
| PPARG | Regulates inflammatory signal and mucin production in airway epithelium. | [75] |
| SMAD1 | *Signaling plays a role in mucus cell differentiation in airway epithelium* | [76] |
| STAT1 | *Enhances PD-L1 expression via IFN signaling* | [72] |
| TGFB1 | *Epithelial derived may act to increase viral burden* | [77] |
| TGFB3 | *Airway mucus hyper-secretion through autophagy* | [78] |
| TP53 | *Encodes human P53 have antiviral effects* | [79] |
| VIM | *EMT marker* | [80] |

gene array we evaluated expression of transcripts associated with a homeostatic/protective function of airway epithelium in naïve and NRG1 treated or untreated RSV infected hBEC; we focused on genes that may be broadly playing roles in (i) epithelial-mesenchymal transition (EMT), cell adhesion, proliferation and epithelial barrier function: [*CDH1*, *CDH2*, *COL1A1*, *CTNNB1*, *VIM*, *ICAM1*, *IGF1R*]; (ii) Transcription factors/transcriptional regulators: [*HIF1A*, *MYC*, *TGFB1*, *TGFB3*, *TP53*, *HDAC1*, *PPARG*, *SMAD1*, *STAT1*]; (iii) cytokine and Immune checkpoint: [*IL6*, *PDCD1*, *CD274* (PD-L1), *CTLA4*] (Fig 7B, 7C and Table 1). Administration of NRG1 to uninfected epithelial cells had little effect on the expression of most genes examined. However, infection with RSV led to a marked, significant upregulation in the expression of many genes. Importantly, treatment with NRG1 reduced expression of nine of these induced genes – *IL6*, *PDCD1*, *STAT1*, *CD274* (PD-L1)*, CTLA-4, HIF1α*, *MYC*, *MCL1*, and *ICAM-1* (Fig 7B). Collectively the expression changes with NRG1 in RSV infected hBEC suggest that NRG1 may regulate genes normally overexpressed with viral infection, potentially helping to maintain epithelial homeostasis and repair, for example by downregulating genes known to be upregulated during necroptosis, such as *PD-L1* and *HIF1α* [51,52]. Further studies are needed to determine if NRG1 induces this protection through direct regulation or via reduced viral titer and/or necroptosis of airway epithelial cells.

While we have not performed an in-depth study on how inhibition of necroptosis leads to improved survival following lethal viral infection, a recent report has found necroptosis inhibition provides a survival benefit in respiratory viral

infections through inhibition of inflammatory mediators, preventing a loss of alveolar integrity and gas exchange [39]. We explored expression of some mediators of inflammation such as *IL6, STAT1, PD-L1,* and found reduced expression in NRG1 treated hBEC following RSV infection (Fig 7B, 7C and Table 1), as well as markedly reduced inflammation in surviving mice (S1 Fig). Moreover, in our study NRG1 treatment reduced airway epithelial leak both *in vitro* (RSV infected hBEC) and *in vivo* (Sendai virus infected mice) suggesting that survival due to necroptosis inhibition potentially is due to improved epithelial barrier function via regulation of mediators of inflammation and homeostasis. Further studies are needed to determine the requirements for each of these gene products in the protection mediated by NRG1.

In conclusion, our studies demonstrate that intranasal administration of NRG1 alone is sufficient to protect mice from a lethal viral infection, as well as reducing airway leak. Administration of NRG1 to epithelial cell cultures *ex vivo* and *in vitro* associated with inhibition of necroptosis and correlated with the reduced fluid leak into the airways *in vivo*. Together, we have identified a novel role for NRG1 in providing protection from a severe respiratory viral infection. However, we acknowledge that other mechanisms besides inhibition of necroptosis could be contributing to the increased survival seen in our studies. Further studies are needed to explore in more detail the mechanism of NRG1 mediated protection, whether NRG1 is the mechanism of atopic protection from respiratory viral induced mortality, and the therapeutic potential of NRG1 for the treatment of severe respiratory viral infections.

## Materials & methods

### Ethics statement

All animal studies were performed under the protocol approved by the Institutional Animal Care and Use Committee of the Abigail Wexner Research Institute at Nationwide Children's Hospital (IACUC number: AR16–00052). Human tissues for cell culture were obtained from de-identified donors. The progenitor cells for these cultures were extracted from de-identified lung tissue provided by The Ohio State University Wexner Medical Center, Comprehensive Transplant Center Human Tissue Biorepository and is exempt from IRB approval.

### Animals

C57BL/6 (wild type) mice were purchased from the Jackson Laboratory (Bar Harbor, Maine, USA) and bred in-house. Unless indicated otherwise mice 8–12 weeks old of both sexes were used for the experiments. No gender difference on outcomes was noted in any of the animal experiments.

### Human air-liquid interphase culture

Human bronchial epithelial cultures (hBEC) progenitors were grown on transwells at the air-liquid interface for 5–6 weeks for the development of pseudostratified well-differentiated airway epithelial layers resembling *in vivo* epithelium with ciliated, goblet and basal cell types as we and others have published [15,81–83].

### Atopic model and SeV infection

C57BL6 mice were sensitized intranasally (i.n.) with 1 μg house dust mite (HDM) extract (catalog no. XPB91D3A2.5; Stallergenes Greer USA, Boston, MA) and one week later challenged for five days with 10 μg HDM extract to make them atopic or were treated with PBS i.n. in a similar manner for non-atopic (NA) controls. Three days after last challenge (HDM or PBS) mice were inoculated i.n. with $2x10^5$ pfu (regular dose) or $2x10^6$ pfu (high dose) SeV [15].

### Airway hyper-reactivity (AHR) and mucus cell measurement

Invasive measurement of AHR was performed by measuring airway resistance to increasing doses of methacholine using the FlexiVent system as we have published [15,84]. Mucus cell metaplasia was determined by performing

Periodic-Acid-Schiff (PAS) staining of formalin fixed lung sections and in a blinded fashion, manually counting and determining the number of PAS+ cells per mm of basement membrane (using ImageJ) as we have reported [15].

## Immunohistochemistry

Paraffin embedded mouse lung tissues were sectioned at 10 µM thickness, dewaxed followed by heat induced antigen retrieval and stained with active caspase-3 antibody (catalog #: AF835-SP, R & D Systems) at 1 µg/ml overnight at 4°C followed by incubation with the anti-rabbit IgG HRP polymer antibody (catalog # VC003, R & D Systems). Tissues were stained with DAB (brown) and counterstained with hematoxylin (blue) following manufacturer's instructions. The number of caspase-3 positive cells per mm of basement membrane were determined as described above for PAS+ cells.

## Western blot analysis

Western blotting was performed on protein lysate from freshly harvested mouse lung tissue using RIPA lysis and extraction buffer (cat# 89901) supplemented with 1x Protease Inhibitor Cocktail (cat# 87786) and 1.5x Phosphatase Inhibitor (cat# 7842) all from ThermoFisher Scientific. Protein lysates (50 µg for each sample) were resolved by SDS-PAGE on 8–16% mini protein gels (cat# 4568104, BioRAD) and transferred on PVDF transfer membranes, 0.2 µm (cat# 88520, ThermoFisher Scientific) to detect specific proteins using antibodies for MLKL (1:1000 of Anti-Mouse M Mlkl [C-term] cat#. 102–12399, RayBiotech), pMLKL (1:1000 of Recombinant Anti-MLKL (phospho S345) antibody cat# ab196436, Abcam) and 1:2000 dilution of GAPDH (14C10) rabbit mAB, cat# 2118) or beta-tubulin (1:1000 of β-Tubulin (D3U1W) Mouse mAb, cat# 86298T) from Cell Signaling as controls. Protein bands were detected by 1:20000 dilution of HRP-conjugated secondary antibodies (goat anti-rabbit-HRP, cat# ab9705, Abcam or goat anti-mouse-HRP cat# 405306, BioLegend). After treating membrane with ultra-sensitive chemiluminescent western blot reagent with horseradish peroxidase enzyme (cat# 34095 Thermo Scientific) for 20–40 seconds protein bands were visualized using the Chemidoc gel imaging system (Bio-RAD, USA). Protein band intensities were quantified using Image Lab software (Bio-Rad, USA). Target protein band intensities in each sample were normalized to GAPDH (for MLKL) or β-tubulin (for p-MLKL) as control for loading variations. Target protein band intensities for all samples were relative to the SeV-alone group. Generation of graphs and statistical analyses (unpaired Student's t-test) were performed in GraphPad Prism 10 with P values of $<0.05$ were considered significant and are indicated in Fig 4D.

## NRG1 ELISA

Detection of NRG1 levels was performed using mouse NRG1 ELISA kit (cat# EKN47308, Biomatik, Delaware USA) following manufacturer's instructions. Freshly prepared standards, tissue extract and cell supernatant were used for the assay and results quantified by plotting against the standard curve with detection range of 15.6-1000 pg/ml and reading O.D. at 450 nm.

## NRG1 exogenous administration and cell culture treatment

Recombinant mouse neuregulin-1/NRG1 protein (carrier free. cat#: 9875-NR, Novus Biologicals, CO, USA) at varying concentrations (10–1000 ng) was given i.n. daily for 5 days before inoculating with high dose SeV. Data were recorded for weight change and survival post infection.

Cell culture basal media was either supplemented with recombinant human NRG1α (cat#: NBP2–35093, Novus Biologicals) or applied on the apical side of hBEC on day 5 and 3 before, and at the time of infection (day 0), with 4000 pfu of rgRSV. Virus inoculation occurred in 100 µl DMEM on the apical side for 4 hr at 37°C in 5% $CO_2$ incubator. GFP levels were quantified by imaging with EVOS cell imaging system at 10x magnification and mean fluorescent intensity determined with ImageJ software [85]. Cell cultures were harvested after 48 h for RNA isolation and qRT-PCR. The rgRSV that encodes *GFP*, and thus, expresses GFP in productively infected cells, was developed by our group [86].

## Vascular leak and epithelial permeability quantification

Vascular leak and alveolar permeability were assessed by measuring accumulation of Evans blue dye (EBD) in the lung or bronchoalveolar lavage (BAL) respectively with modifications as described by our group and others [34,35,87]. Briefly, at day 5 or 8 PI SeV, 100 μL EBD (cat# E2129, Sigma-Aldrich) at 20 mg/kg was injected i.v. into the tail vain. One hour later BAL was performed and then lungs harvested. To lung samples formamide was added to extract EBD while BAL's EBD was extracted without formamide treatment. The extracted dye was quantitated by spectrophotometry as we have published, measuring absorbance at 620 nm with absorbance at 740 nm used as a baseline control [35].

## Epithelial leak assay in hBEC

To determine airway epithelial permeability fluorescein isothiocyanate (FITC)–dextran (mol wt 70,000, cat# 46945, Millipore Sigma) 1mg/ml was dissolved in 1x PBS (without calcium and magnesium) and 25 μl was applied on the apical side of the transwells after rinsing the surface with DMEM. Basal media was collected at various time intervals post RSV inoculation and fluorescence measured at 530 nm to determine epithelial leak. NRG1 (100 ng) was applied to the apical or basal media as outlined above (see section on cell culture treatment); necrosulfonamide (NSA) (cat# S8251, Selleckchem) 5 μM was applied to the basal media 3 days before, and at the time of inoculation (day 0), with 4000 pfu of rgRSV.

## Apoptosis and secondary necrosis assay

Real-time non-lytic assay to differentiate secondary necrosis occurring during late-phase apoptosis (annexin V binding) was performed using using RealTime Annexin V Apoptosis and Necrosis Assay kit (cat# JA1011, Promega). LET-1 cells ($6.0x10^4$) were plated 24 hours before the assay with or without NRG1 (500 ng) in white opaque 96-well plates. After 24h the cells were treated with SeV (9000 pfu) in respective wells with or without NRG1. Annexin V smBiT and LgBiT proteins were added along with the NanoBiT substrate and the necrosis reagent as per the manufacturer's protocol. Plate was read at 470–10nm in the luminescence channel for apoptosis and in the FITC fluorescence channel (515/30 nm) for necrosis every hour for 24 hours using BMG Labtech's Clariostar plate reader kept at $37^0$C with 5% $CO_2$.

## RNA sequencing

Cells from whole mouse lungs were harvested and flow cytometry and Fluorescence-Activated Cell Sorting (FACS) performed as we previously reported using standard cell staining techniques with CD11c antibody (Clone N418; catalog no. 12-0114-82, ThermoFisher) or Armenian Hamster IgG Isotype Control (Clone: eBio299Arm, catalog no. 12-4888-81, ThermoFisher), [15,16,88]. Sorted cells were >85% CD11c+ (S3 Fig). Whole transcriptome profiling was performed by preparing strand-specific RNA-seq libraries using NEB Next Ultra II Directional RNA Library Prep Kit for Illumina, following the manufacturer's recommendations. In summary, total RNA was assessed using RNA 6000 Nano kit on Agilent 2100 Bioanalyzer (Agilent Biotechnologies) and Qubit RNA HS assay kit (Life Technologies). A 140–500 ng aliquot of total RNA was rRNA depleted using NEB's Human/Mouse/Rat RNAse-H based Depletion kit (New England BioLabs). Following rRNA removal, mRNA was fragmented and then used for first- and second-strand cDNA synthesis with random hexamer primers and ds cDNA fragments undergoing end-repair and a-tailing and ligation to dual-unique adapters (Integrated DNA Technologies). Adaptor-ligated cDNA was amplified by limit-cycle PCR. Library quality was analyzed on Tapestation High-Sensitivity D1000 ScreenTape (Agilent Biotechnologies) and quantified by KAPA qPCR (KAPA BioSystems). Libraries were pooled and sequenced at 2 x 150 bp read lengths on the Illumina HiSeq 4000 platform to generate approximately 60–80 million paired-end reads per sample. Differential expression analysis was performed and significant differentially expressed features between the two groups with an absolute value of fold change ≥ 1.5 and an adjusted p-value of ≤ 0.10 (10% FDR) were recorded.

### RNA isolation and quantitative real-time PCR (qRT-PCR)

Total RNA from hBEC was isolated using mirVana miRNA and total RNA isolation kit (catalog no. AM1560, Thermo Fisher Scientific). For qRT-PCR cDNA was synthesized using Maxima H Minus cDNA synthesis kit (catalog no. M1681; Thermo Fisher Scientific). PCR primers assay plates were custom made with validated primers for use with EvaGreen dye-based chemistry (cat# PrimePCR 10025218, Bio-RAD, USA). Samples were run on CFX96 Touch Real-Time Detection System (Bio-RAD, USA). $C_T$ values normalized with *GAPDH* or *MRPL13* and expressed as fold change relative to naïve control. For the array we selected 24 genes that broadly fall into three groups: i) epithelial to mesenchymal transition signature genes; (ii) genes in RNA virus infection panel of pre-designed human PrimePCR by BioRad; (iii) fibroblast genes previously reported to be regulated by NRG1 [50].

The following human genes were included in the assay: *BAX* (BCL2-associated X protein); *BCL2L1* (BCL2-like 1); *CD274* (PD-L1); *CDH1* (cadherin 1, type 1, E-cadherin (epithelial); *CDH2* (cadherin 2, type 1, N-cadherin (neuronal); *COL1A1* (collagen, type I, alpha 1); *CTLA4* (ICOS, cytotoxic T-lymphocyte-associated protein 4); *CTNNB1* (catenin (cadherin-associated protein), beta 1, 88kDa), *GAPDH* (glyceraldehyde-3-phosphate dehydrogenase); *HDAC1* (histone deacetylase 1); *HIF1A* (hypoxia inducible factor 1); *ICAM1* (intercellular adhesion molecule 1); *IGF1R* (insulin-like growth factor 1 receptor); *IL6* (interleukin 6); *MCL1* (myeloid cell leukemia sequence 1 (BCL2-related)); *MRPL13* (mitochondrial ribosomal protein L13); *MYC* (v-myc myelocytomatosis viral oncogene homolog (avian); *PDCD1* (programmed cell death 1); *PPARG* (peroxisome proliferator-activated receptor gamma); *SMAD1* (SMAD family member 1); *STAT1* (signal transducer and activator of transcription 1, 91kDa); *TGFB1* (transforming growth factor, beta 1); *TGFB3* (transforming growth factor, beta 3); *TP53* (tumor protein p53); *VIM* (vimentin).

### Statistical analysis

All statistical analyses were performed using Prism 9 (GraphPad Software Inc.). All normally distributed data are presented as mean ± SEM with survival analyses using Kaplan-Meier curves and log-rank (Mantel-Cox) tests; non-normally distributed data are shown as median ± inter-quartile range (IQR). Student's *t* test (for normally distributed data) or Mann-Whitney U (for non-normally distributed data) was used to assess significant differences between two means. For all tests, $p < 0.05$ was considered statistically significant.

### Supporting information

**S1 Table. Differentially expressed genes in atopic vs non-atopic CD11c$^+$ cells.** Differential gene expression (DGE) in lung CD11c$^+$ cells, harvested from mice made atopic with house dust mite (HDM) extract relative to non-atopic [PBS control] mice (on day 14 post initial HDM/PBS sensitization).
(XLSX)

**S1 Fig. NRG1 treatment does not alter granulocyte numbers but markedly reduces lung inflammation. (A)** Granulocyte numbers determined from flow cytometric analysis based on forward and side scatter demonstrating no significant difference in high dose SeV infected mice treated with PBS or NRG1 as in Fig 2A at d0, d5, d7 PI SeV (n = 3). **(B)** Hematoxylin and eosin (H&E) staining of paraffin fixed mouse lung comparing high dose SeV (2x10$^6$ pfu) infected and NRG1 treated (NRG-1-SeV) mice with regular dose SeV (2x10$^5$ pfu) infected but otherwise untreated mice at day 21 post infection showing markedly reduced inflammation in the NRG1 treated group even with the higher viral dose. Images taken at 5x magnification, n = 3 per group.
(TIF)

**S2 Fig. Atopic mice demonstrate reduced airway fluid leak. (A)** Ratio of EBD in the airway (BAL) to lung at day 5 post inoculation (PI) high dose SeV (2x10$^6$ pfu) is lower for mice made atopic (HHS, red circles) compared to non-atopic mice

(PPS, black circles). **(B)** Similar to (A) but using regular dose SeV ($2x10^5$ pfu) and measuring EBD at day 8 PI. For (A) and (B) median $\pm$ IQR shown, Mann-Whitney U test, n $\geq$ 5.
(TIF)

**S3 Fig. CD11c$^+$ cell purity.** FACS of CD11c$^+$ cells from lung stained with PE conjugated CD11c antibody shows gating strategy which led to $\geq$ 87% purity of the selected population (left panel: FSC vs SSC, middle panel: singlets, right panel: CD11c$^+$ population).
(TIF)

## Acknowledgments

The authors thank the Genomic Services at The Abigail Wexner Research Institute (AWRI) for performing RNAseq. Production of the hBEC cultures was performed by the Epithelial Production, Analysis, and Innovation Core (EPAIC) at Nationwide Children's Hospital (NCH) and The Ohio State University (OSU), which are supported by the Cystic Fibrosis Foundation (CFF) through the Columbus Cures Cystic Fibrosis (C3) Research Development Program Grant (MCCOY17R2, and MCCOY19Ro), and the OSU Center for Clinical and Translational Science (UL1TR002733). Source tissues for hBEC cultures were provided by the Comprehensive Transplant Center Human Tissue Biorepository of The OSU Wexner Medical Center. Further, we want to thank Dr. Israel Cotzomi-Ortega of AWRI for providing technical expertise in conducting some experiments and Joshua Walum of AWRI for help with data repository submission.

## Author contributions

**Conceptualization:** Syed-Rehan A Hussain, Mitchell H Grayson.

**Formal analysis:** Syed-Rehan A Hussain, Michelle Rohlfing, Jennifer Santoro, Kaushik Muralidharan, Mathew S Bochter, Mitchell H Grayson.

**Funding acquisition:** Mark E Peeples, Mitchell H Grayson.

**Investigation:** Syed-Rehan A Hussain, Michelle Rohlfing, Jennifer Santoro, Mathew S Bochter, Mitchell H Grayson.

**Methodology:** Syed-Rehan A Hussain, Michelle Rohlfing, Jennifer Santoro, Phylip Chen, Kaushik Muralidharan, Mark E Peeples.

**Project administration:** Mitchell H Grayson.

**Resources:** Phylip Chen, Mark E Peeples, Mitchell H Grayson.

**Supervision:** Mitchell H Grayson.

**Validation:** Syed-Rehan A Hussain, Mitchell H Grayson.

**Writing – original draft:** Syed-Rehan A Hussain, Mitchell H Grayson.

**Writing – review & editing:** Syed-Rehan A Hussain, Mark E Peeples, Mitchell H Grayson.

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
