## [Decision Letter · Decision Letter 0]

8 Dec 2024

PPATHOGENS-D-24-02280

Neuregulin-1 inhibits mouse respiratory virus induced necroptosis and death

PLOS Pathogens

Dear Dr. Hussain,

Thank you for submitting your manuscript to PLOS Pathogens. After careful consideration, we feel that it has merit but does not fully meet PLOS Pathogens's publication criteria as it currently stands. Therefore, we invite you to submit a revised version of the manuscript that addresses the points raised during the review process.

Please submit your revised manuscript within 60 days Feb 06 2025 11:59PM. If you will need more time than this to complete your revisions, please reply to this message or contact the journal office at plospathogens@plos.org. Please include the following items when submitting your revised manuscript:

We look forward to receiving your revised manuscript.

Kind regards,

Makoto Takeda

Academic Editor

PLOS Pathogens

Benhur Lee

Section Editor

PLOS Pathogens

Michael Malim

Editor-in-Chief

PLOS Pathogens

orcid.org/0000-0002-7699-2064

**Additional Editor Comments :**

This manuscript has been reviewed by three experts. As a result, some reviewers pointed out that the association between the expression level of NPG1 and its effect on mitigating disease severity has not been sufficiently demonstrated. Additionally, one reviewer has raised questions regarding the novelty of the study. On the other hand, another reviewer acknowledged the novelty of this research and evaluated it as a significant contribution to the field. Please carefully consider all reviewers' comments and provide thorough responses accordingly.

**Journal Requirements:**

At this stage, the following Authors/Authors require contributions: Mathew S Bochter. Please ensure that the full contributions of each author are acknowledged in the "Add/Edit/Remove Authors" section of our submission form.

2) We ask that a manuscript source file is provided at Revision. Please upload your manuscript file as a .doc, .docx, .rtf or .tex. If you are providing a .tex file, please upload it under the item type LaTeX Source File and leave your .pdf version as the item type Manuscript.

https://journals.plos.org/plospathogens/s/submission-guidelines#loc-parts-of-a-submission

4) We noticed that you used the phrase 'not shown' in the manuscript. We do not allow these references, as the PLOS data access policy requires that all data be either published with the manuscript or made available in a publicly accessible database. Please amend the supplementary material to include the referenced data or remove the references.

5) We do not publish any copyright or trademark symbols that usually accompany proprietary names, eg ©,  ®, or TM  (e.g. next to drug or reagent names). Therefore please remove all instances of trademark/copyright symbols throughout the text, including:

- ® on page: 18

- TM on pages: 18, 21, and 23.

6) Please upload all main figures as separate Figure files in .tif or .eps format. For more information about how to convert and format your figure files please see our guidelines: 

7) Please upload a copy of Figure 4E which you refer to in your text on page 12. Or, if the figure is no longer to be included as part of the submission please remove all reference to it within the text.

8)  Please label the table and amend its reference accordingly. 

9) We have noticed that there is a reference to supplemental figure 2 on page 19. However, there is no corresponding file uploaded to the submission. Please upload it as a separate file with the item type 'Supporting Information'.

10) We have noticed that you have uploaded Supporting Information files, but you have not included a list of legends. Please add a full list of legends for your Supporting Information files after the references list.

11) Thank you for stating that " I have deposited our RNA seq data, relevant to the submitted manuscript (PPATHOGENS-D-24-02280) to GEO repository following is the link and accession no

https://www.ncbi.nlm.nih.gov/geo/query/acc.cgi

GEO accession: GSE281143" . We  noticed that the data is currently private and is scheduled to be released on Nov 04, 2025. We strongly recommend all authors deposit their data before acceptance, as the process can be lengthy and hold up publication timelines. Please note that, though access restrictions are acceptable now, your entire minimal dataset will need to be made freely accessible if your manuscript is accepted for publication. This policy applies to all data except where public deposition would breach compliance with the protocol approved by your research ethics board. If you are unable to adhere to our open data policy, please kindly revise your statement to explain your reasoning and we will seek the editor's input on an exemption.

12) Please amend your detailed Financial Disclosure statement. This is published with the article. It must therefore be completed in full sentences and contain the exact wording you wish to be published.

13) Please ensure that the funders and grant numbers match between the Financial Disclosure field and the Funding Information tab in your submission form. Note that the funders must be provided in the same order in both places as well. Currently, this funding information " the Robert & Edgar Wolfe Foundation (to MHG) " is missing from the Funding Information tab.

Please indicate by return email the full and correct funding information for your study and confirm the order in which funding contributions should appear. Please be sure to indicate whether the funders played any role in the study design, data collection and analysis, decision to publish, or preparation of the manuscript.

**Reviewers' Comments:**

Reviewer's Responses to Questions

**Part I - Summary**

Reviewer #1: 1. Lack of Novelty: The findings on the protective effect of atopy in viral infections lack novelty, as similar protective mechanisms have been demonstrated in previous studies using influenza virus in mouse models. Studies have already shown that pre-existing atopy can reduce disease severity in influenza infections through mechanisms such as NK cell activation and Type III interferon induction

2. The authors challenged C57BL6 mice intranasally with 10 μg of house dust mite (HDM) extract 3 days before SeV infection, but they have not ruled out the possibility that the HDM extract could induce nonspecific innate immunity, including interferons, which may improve the survival rate of virus-infected mice or suppress SeV replication.

3. Rational: Despite the RNAseq analysis results showing significant differences in the expression levels of various genes, including Adam11, between NA mice and atopic mice, the rationale for focusing on neuregulin-1 is insufficient.

4. Limited Mechanistic Insight: The manuscript does not provide sufficient mechanistic insight into how neuregulin-1 (NRG1) mediates protection against respiratory viral infections. While the authors highlight NRG1’s role in reducing necroptosis, they do not adequately demonstrate how this leads to the observed survival benefit, leaving gaps in the mechanistic understanding.

5. Overemphasis on Correlative Data: Much of the evidence provided in the study appears to be correlational, particularly regarding the association between reduced necroptosis and improved survival. Stronger causal data, such as using knockout models, could strengthen the conclusions.

Reviewer #2: In this manuscript, Hussain SA et al. demonstrated the effect of pre-existing atopy on respiratory virus infection. The authors found that increased NRG1 protects mice from SeV infection via upregulation of repair process. This manuscript presents novel findings, but the results appear somewhat disconnected and lack a cohesive conclusion.

Reviewer #3: Following up on previous work by the authors describing reduced severity of respiratory viral infections in mice made atopic, this study reports upregulation of NRG1 in atopic mice. In a series of in vivo and in vitro experiments, NRG1 is shown to mediate increased survival of animals in a SeV mouse infection model through reduced alveolar leakage and inhibition of virus-induced necroptosis.

These results advance the mechanistic understanding of an unexpected phenotype and identify determinants of disease severity. Data are well presented and overall support the conclusions drawn. Although no direct causality between NRG1 expression levels and greater protection of atopic mice has been established, the authors recognize and appropriately discuss possible limitations of the work. Thorough final editing of the text is recommended to address a few inaccuracies (i.e. reporter gene expression is not equivalent to virus titer (line 256)), but overall this is an important study that will advance the field.

**Part II – Major Issues: Key Experiments Required for Acceptance**

Reviewer #1: (No Response)

Reviewer #2: 1. In line 87: “our high-fidelity model of respiratory viral infection utilizes a natural rodent pathogen, SeV…”, SeV infection cannot be an animal model for RSV infection in humans. It is better to reorganize this section.

2. Fig 1B: please provide MLD50 of SeV infection. How much mice are survived from the “regular” dose infection?

3. The authors need to show the severity of asthma in this model. In this model, there is no PAS+ cells as shown in Fig 2E, indicating that the asthma symptoms are very weak. Please examine levels of IgE and granulocytes. Is this a chronic inflammation model?

4. In line 161: “In a dose responsive fashion”, there is no dose response.

5. Fig 3: EBD should be examined in SeV-infected atopic model.

6. Fig 2C: The viral titer does not drop in NRG1 treated mice, even though it does in the atopic model.

7. Fig 5A: The results are very unclear, the authors have to provide WB results with similar lysate amount.

8. It is unclear why necroptosis is inhibited by NRG1 treatment.

9. Is the epithelial leak regulated by necroptosis?

Reviewer #3: (No Response)

**Part III – Minor Issues: Editorial and Data Presentation Modifications**

Reviewer #1: (No Response)

Reviewer #2: (No Response)

Reviewer #3: (No Response)

PLOS authors have the option to publish the peer review history of their article (what does this mean? ). If published, this will include your full peer review and any attached files.

**Do you want your identity to be public for this peer review?** For information about this choice, including consent withdrawal, please see our Privacy Policy .

Reviewer #1: No

Reviewer #2: No

Reviewer #3: No

**Figure resubmission:**
---

## [Decision Letter · Decision Letter 1]

26 Mar 2025

PPATHOGENS-D-24-02280R1

Neuregulin-1 inhibits mouse respiratory virus induced necroptosis and death

PLOS Pathogens

Dear Dr. Grayson,

Thank you for submitting your manuscript to PLOS Pathogens. After careful consideration, we feel that it has merit but does not fully meet PLOS Pathogens's publication criteria as it currently stands. Therefore, we invite you to submit a revised version of the manuscript that addresses the points raised during the review process.

Please pay attention to the "**Additional Editor Comments"** below and respond to reviewer #2 remarks regarding whether the model presented represents necroptosis. These comments are not something that can be addressed by additional experimentation but perhaps additional explication regarding the limitations of the model and/or a change in title is warranted. 

Please submit your revised manuscript within 30 days May 25 2025 11:59PM. If you will need more time than this to complete your revisions, please reply to this message or contact the journal office at plospathogens@plos.org. Please include the following items when submitting your revised manuscript:

We look forward to receiving your revised manuscript.

Kind regards,

Makoto Takeda

Academic Editor

PLOS Pathogens

Benhur Lee

Section Editor

PLOS Pathogens

Sumita Bhaduri-McIntosh

Editor-in-Chief

PLOS Pathogens

orcid.org/0000-0003-2946-9497

Michael Malim

Editor-in-Chief

PLOS Pathogens

orcid.org/0000-0002-7699-2064

**Additional Editor Comments:**

According to the concerns one of the reviewers raised, it is unclear what pathological condition this model reflects. In other words, it is difficult to determine whether this model has aspects of an atopic model or whether the inflammation-inducing model using HDM is merely a trigger for discovering NRG1. Another particularly important point is whether the observed immune and inflammatory response, and pathological features are completely consistent with necroptosis. This is because the title of this manuscript is based on the claim that NRG1 inhibits necroptosis, so a satisfactory explanation of these points is required. Is it possible to respond appropriately to the reviewer's concern?

**Journal Requirements:**

At this stage, the following Authors/Authors require contributions: Mitchell H Grayson. Please ensure that the full contributions of each author are acknowledged in the "Add/Edit/Remove Authors" section of our submission form.

2) Please ensure that the funders and grant numbers match between the Financial Disclosure field and the Funding Information tab in your submission form. Note that the funders must be provided in the same order in both places as well.

**Reviewers' Comments:**

Reviewer's Responses to Questions

**Part I - Summary**

Reviewer #1: (No Response)

Reviewer #2: Although HDM was administered, this is not an atopic model but rather an experiment in which mice were infected with artificially induced acute inflammation, not chronic inflammation, and there are discrepancies in the interpretation of the results. Necroptosis, which is an inflammatory cell death, was suppressed by NRG1, but inflammatory responses such as granulocyte recruitment were not suppressed. This suggests that necroptosis is not relevant to this mouse model. The results are not consistent and do not respond to this reviewer's concerns at all. Also, the data could not support the interpretation of the results.

Reviewer #3: My concerns have been addressed. This is a well-executed study that will advance the field.

**Part II – Major Issues: Key Experiments Required for Acceptance**

Reviewer #1: (No Response)

Reviewer #2: (No Response)

Reviewer #3: (No Response)

**Part III – Minor Issues: Editorial and Data Presentation Modifications**

Reviewer #1: (No Response)

Reviewer #2: (No Response)

Reviewer #3: (No Response)

PLOS authors have the option to publish the peer review history of their article (what does this mean? ). If published, this will include your full peer review and any attached files.

**Do you want your identity to be public for this peer review?** For information about this choice, including consent withdrawal, please see our Privacy Policy .

Reviewer #1: No

Reviewer #2: No

Reviewer #3: No

**Figure resubmission:**
---

## [Editor Report · Decision Letter 2]

11 Apr 2025

Dear Dr Grayson,

We are pleased to inform you that your manuscript 'Neuregulin-1 prevents death from a normally lethal respiratory viral infection' has been provisionally accepted for publication in PLOS Pathogens.

Best regards,

Makoto Takeda

Academic Editor

PLOS Pathogens

Benhur Lee

Section Editor

PLOS Pathogens

Sumita Bhaduri-McIntosh

Editor-in-Chief

PLOS Pathogens

orcid.org/0000-0003-2946-9497

Michael Malim

Editor-in-Chief

PLOS Pathogens

orcid.org/0000-0002-7699-2064
---

## [Editor Report · Acceptance letter]

Dear Dr Grayson,

We are delighted to inform you that your manuscript, "Neuregulin-1 prevents death from a normally lethal respiratory viral infection," has been formally accepted for publication in PLOS Pathogens.

Best regards,

Sumita Bhaduri-McIntosh

Editor-in-Chief

PLOS Pathogens

orcid.org/0000-0003-2946-9497

Michael Malim

Editor-in-Chief

PLOS Pathogens

orcid.org/0000-0002-7699-2064